# Cannabidiol Reduces Short- and Long-Term High Glutamate Release after Severe Traumatic Brain Injury and Improves Functional Recovery

**DOI:** 10.3390/pharmaceutics14081609

**Published:** 2022-08-02

**Authors:** Cindy Santiago-Castañeda, Saúl Huerta de la Cruz, Christopher Martínez-Aguirre, Sandra Adela Orozco-Suárez, Luisa Rocha

**Affiliations:** 1Department of Pharmacobiology, Center for Research and Advanced Studies (CINVESTAV), Mexico City 14330, Mexico; cindy.santiago@cinvestav.mx (C.S.-C.); saul.huerta@cinvestav.mx (S.H.d.l.C.); christopher_mtz@cinvestav.mx (C.M.-A.); 2Unit for Medical Research in Neurological Diseases, Specialties Hospital, National Medical Center SXXI (CMN-SXXI), Mexico City 06720, Mexico; sorozco5@hotmail.com

**Keywords:** traumatic brain injury, cannabidiol, glutamate, sensorimotor function, body weight, mortality

## Abstract

This study aimed to determine if orally administered cannabidiol (CBD) lessens the cortical over-release of glutamate induced by a severe traumatic brain injury (TBI) and facilitates functional recovery. The short-term experiment focused on identifying the optimal oral pretreatment of CBD. Male Wistar rats were pretreated with oral administration of CBD (50, 100, or 200 mg/kg) daily for 7 days. Then, extracellular glutamate concentration was estimated by cortical microdialysis before and immediately after a severe TBI. The long-term experiment focused on evaluating the effect of the optimal treatment of CBD (pre- vs. pre- and post-TBI) 30 days after trauma. Sensorimotor function, body weight, and mortality rate were evaluated. In the short term, TBI induced a high release of glutamate (738% ± 173%; *p* < 0.001 vs. basal). Oral pretreatment with CBD at all doses tested reduced glutamate concentration but with higher potency at when animals received 100 mg/kg (222 ± 33%, *p* < 0.01 vs. TBI), an effect associated with a lower mortality rate (22%, *p* < 0.001 vs. TBI). In the long-term experiment, the TBI group showed a high glutamate concentration (149% *p* < 0.01 vs. SHAM). In contrast, animals receiving the optimal treatment of CBD (pre- and pre/post-TBI) showed glutamate concentrations like the SHAM group (*p* > 0.05). This effect was associated with high sensorimotor function improvement. CBD pretreatment, but not pre-/post-treatment, induced a higher body weight gain (39% ± 2.7%, *p* < 0.01 vs. TBI) and lower mortality rate (22%, *p* < 0.01 vs. TBI). These results support that orally administered CBD reduces short- and long-term TBI-induced excitotoxicity and facilitated functional recovery. Indeed, pretreatment with CBD was sufficient to lessen the adverse sequelae of TBI.

## 1. Introduction

Traumatic brain injury (TBI) originates from an external force, such as a blow, jolt, or penetrating head injury, which causes temporary or permanent impairment of brain functions [1]. About 40% of patients with moderate–severe TBI may have permanent deficits and disabilities. TBI is the leading cause of death in people under 45 years [2,3]. TBI requires hospitalization and rehabilitation, resulting in high hospital expenses [1,4,5]. Costs to treat a patient with a severe TBI exceed 400 billion USD annually, depending on the severity of the trauma and previous comorbidities [6,7].

The physiopathology of TBI involves a primary injury that is immediate and occurs after mechanical damage [8,9]. Subsequently, secondary injury is associated with activating biochemical cascades that lead to metabolic and cellular changes [10]. Both primary and secondary injuries are associated with increased glutamatergic neurotransmitters [11]. Furthermore, high extracellular levels of glutamate in the brain (up to 120 h after TBI) predict a poor prognosis in TBI patients [12]. At high concentrations, glutamate promotes excitotoxicity [13,14], neuroinflammation [15,16], oxidative stress [17], and cell death [18]. Maintaining increased glutamatergic neurotransmission after a severe TBI represents a risk factor for developing disorders such as Parkinson’s disease [19], Alzheimer’s disease [20], and post-traumatic epilepsy [21]. It is important to mention that, at present, there are still no therapeutic strategies that prevent the long-term consequences of TBI.

Cannabidiol (CBD) is a molecule of growing interest in different conditions such as pain [22], drug-resistant epilepsy [23], and depression [24]. CBD induces protective effects with a wide therapeutic window when applied systemically or orally [25,26]. Orally administered CBD presents a bioavailability of 6–13% that increases when co-administered with high-fat food [27,28]. The maximum concentration in plasma (C_max_) by oral administration is achieved at 3 h and is influenced by the dose applied [28]. CBD metabolism is modulated by CYP 450 enzymes (3A4 and C219) [29]. At present, oral treatment with CBD (Epidiolex) is used for the treatment of epilepsies that are difficult to control [30].

On the other hand, experimental evidence supports that intracerebroventricular pretreatment with CBD reduces the volume of brain injury, neuroinflammation, and blood–brain barrier (BBB) damage induced by brain hypoxia/ischemia [31]. The protective effects induced by CBD pretreatment can be explained because of diminished over-release of glutamate [32] and a reduction in proinflammatory cytokine expression [33], oxidative stress [34], and neurodegeneration [35]. Concerning TBI, it is described that systemic administration of CBD after trauma results in decreased injury [36]. Studies indicate that orally administered CBD after TBI reduces allodynia and neurological dysfunctions [37]. This group of evidence suggests that orally pretreatment with CBD reduces the TBI-induced consequences. However, it is unknown if orally administered CBD before severe TBI prevents long-term glutamate-induced excitotoxicity and the functional deficits. The present study aimed to determine whether oral pretreatment with CBD prevents high glutamate concentration in the cortex and reduces short- and long-term sensorimotor impairment, body weight loss, and mortality after severe TBI in rats. We also investigated if the effects of CBD were augmented when applied pre- and post-TBI.

## 2. Materials and Methods

Two experiments were carried out: (a) a short-term experiment to identify the optimal dose of CBD to reduce the TBI-induced acute glutamate over-release, and (b) a long-term experiment to analyze the effects of pre-TBI and pre-TBI-post treatment with the optimal dose of CBD on glutamate over-release and functional deficits induced long-term after a severe TBI. Purified hemp oil extract for oral administration (donated by HempMeds PX, Lindon, UT, USA) was used for the present study. It contained 99.35% CBD and 0.65% cannabidivarin (CBDV).

### 2.1. Animals

Male Wistar rats weighing between 250 and 300 g were used. The animals were kept in transparent acrylic boxes under controlled conditions (12 h light/dark cycles, 22–25 °C), with access to food and water ad libitum. The experimental protocol was performed following the Mexican Official Standard (NOM-062ZOO-1999) and the Ethics Committee of the Center for Research and Advanced Studies of the National Polytechnic Institute (Protocol CICUAL 0326-22 with approval date 4 May 2022).

### 2.2. Stereotaxic Surgery

Rats were anesthetized with ketamine (80 mg/kg, i.p.) and xylazine (17 mg/kg, i.m.), and then placed on a stereotaxic frame. A 1.5 cm sagittal incision was made in the midline between the ears extending toward the nose, and the periosteum was removed. A 5 mm diameter craniotomy (−5 mm anteroposterior; 4.0 mm lateral; 4.0 mm height) [38] was performed, keeping the dura intact [39]. A female Luer lock was then fixed with 3 M Vetbond surgical glue (Deutschland GmbH, Germany). For the short-term experiment, the Luer lock was covered with sterile silicone foam and removed immediately before the TBI induction. A stainless-steel screw was implanted anterior and lateral to the bregma and fixed to the skull with dental acrylic.

For the microdialysis experiments, a stainless-steel guide cannula was implanted near the craniotomy (−0.5 mm anteroposterior; 2.2 mm lateral; 1.5 mm height) [38]. In addition, four stainless-steel screws were implanted anterior to bregma (left and right), lateral, and posterior to the craniotomy. After surgery, animals were left to recover for 7 days.

### 2.3. Induction of Severe TBI

In the short-term experiment, animals were sedated with isoflurane 3% during the microdialysis experiment. Then, the sterile silicone foam covering the Luer lock was removed, and the craniotomy was exposed. A lateral fluid percussion injury (LFPI) was induced through the craniotomy using a fluid percussion device (AmScien Instruments, Model FP 302, Richmond, VA, USA) [39]. In the long-term experiments, animals pretreated with CBD or vehicle were subjected to a severe TBI 90 min after induction of anesthesia for the stereotaxic surgery (Figure 1). The acrylic helmet of these animals was removed immediately after TBI induction, and the skin was sutured. Rats received tramadol administration (20 mg/kg, s.c. NorVet) 15 min after TBI and were kept under observation in heated beds for 3 h. Then, they were allowed to recover for 7 days. The pressure pulse’s strength for short- and long-term experiments was in the range of 2.6 to 3.3 atm to induce severe TBI [40]. Animals that lost more than 30% of their initial body weight were discarded for further evaluation.

### 2.4. Groups for the Short-Term Experiment

This experiment was designed to identify the optimal oral pretreatment of CBD that reduces the high extracellular glutamate levels evoked immediately after severe TBI in the cortex. The effects of CBD oral pretreatment on the sensorimotor function, body weight, and mortality 2 days after TBI were also estimated. The effects of orally administered CBD were evaluated in a dose–response manner. According to a previous study from our laboratory, we found that the orally administered CBD at 200 mg/kg was able to reduce the seizure severity induced by 3-mercaptopropionic acid [41]. Hence, for the present study, we decided to evaluate the doses of 50, 100 and 200 mg/kg.

The animals were randomly divided into five experimental groups.

#### 2.4.1. CBD200 + TBI (*n* = 11)

Under general anesthesia, rats underwent a craniotomy and the implantation of a Luer lock for the subsequent induction of severe TBI. In addition, a guide cannula was implanted 1 mm above the left cerebral cortex and 3 mm closer to the Luer lock (see Section 2.2). Seven days after the surgery, the animals received the oral administration of CBD (200 mg/kg) daily for 7 days. One day after the last CBD administration, microdialysis was performed. Initially, perfusates were obtained in the cortex under basal conditions for 2 h. The animals were then sedated with 3% isoflurane, and a LFPI of 3.0 ± 0.14 atm was applied on the craniotomy via the Luer lock implanted. Immediately after induction of severe TBI, continuous dialysates were obtained for 5 h. Then, dialysates were processed to quantify glutamate concentration by high-performance liquid chromatography (HPLC, Waters^®^ model 474, Stoughton, MA, USA). Body weight and sensorimotor function (Neuroscore test) were evaluated 1 day before and 2 days after TBI induction. Animals were sacrificed 5 days after TBI induction (Figure 1). The brain was dissected to verify the implantation of the cannula in the cortex.

#### 2.4.2. CBD100 + TBI (*n* = 9)

Animals were manipulated as described for the CBD200 + TBI group, with the difference that they received the oral administration of CBD at 100 mg/kg. This group received a LFPI of 3.07 ± 0.09 atm to induce a severe TBI.

#### 2.4.3. CBD50 + TBI (*n* = 9)

Rats received the same manipulation as described for the CBD200 + TBI group, except that CBD was administered orally at 50 mg/kg. Animals underwent severe TBI as consequence of applying a LFPI of 3.1 ± 0.14 atm.

#### 2.4.4. TBI (*n* = 14)

Animals were manipulated as described above (CBD200 + TBI group), with the difference being that coconut oil (vehicle) was administered orally (5 mL/kg) instead of CBD. They received a LFPI of 2.97 ± 0.11 atm to induce a severe TBI.

#### 2.4.5. SHAM (*n* = 7)

Rats were manipulated similarly to the TBI group, except that LFPI was not induced.

### 2.5. Groups for the Long-Term Experiment

This experiment was designed to determine whether the oral administration of CBD applied pre- and post-LFPI reduces the over-release of glutamate and facilitates the functional recovery long-term after severe TBI. For this experiment, animals received the dose of CBD that induced the optimal effect to prevent elevated extracellular glutamate release in the short-term experiment.

#### 2.5.1. CBD + TBI + CBD (*n* = 10)

The animals received the optimal oral administration of CBD for 7 days (see short-term experiment). One day after the last CBD administration and under general anesthesia, the animals underwent severe TBI as a consequence of LFPI of 3.02 ± 0.09 atm. Subsequently, rats received daily oral CBD administration for another 7 days after TBI. Twenty-three days after trauma, the animals were anesthetized and implanted with a guide cannula on cerebral cortex 3 mm closer to the craniotomy (see Section 2.2). Seven days after surgery, microdialysis was performed to determine the extracellular concentrations of glutamate for 3 h. Dialysates obtained from this procedure were analyzed as described for the short-term experiment. The body weight and sensorimotor function (Neuroscore test) were estimated 1 day before and 2, 7, 14, 21, and 28 days post-TBI. (Figure 1). At the end of the experiment, the animals were sacrificed by decapitation, and the brain was dissected to identify the area of cannula implantation.

#### 2.5.2. CBD + TBI (*n* = 9)

Animals were manipulated as described for the CBD + TBI + CBD group, except they received daily oral CBD administration only before TBI induction. These animals received a LFPI of 2.98 ± 0.15 atm.

#### 2.5.3. TBI (*n* = 11)

The experimental procedure for this group was as described for the CBD + TBI group, but animals received daily oral administration of vehicle (5 mL/kg) instead of CBD. Severe TBI was induced with a LFPI of 2.97 ± 0.11 atm.

#### 2.5.4. CBD + SHAM (*n* = 7)

Rats were manipulated as described for the CBD + TBI group, but LFPI was not induced.

#### 2.5.5. SHAM (*n* = 8)

This group was manipulated as described for the CBD + SHAM group, but rats received daily oral administration of vehicle (5 mL/kg) instead of CBD.

### 2.6. Evaluation of Sensorimotor Function with Neuroscore Test

The Neuroscore test was used to estimate the sensorimotor function of the rats. This test consists of four functional evaluations: (1) grasping ability on an inclined plane at different angles (60° to 75°), (2) forelimb and (3) hindlimb extension and counter flexion, and (4) ability to resist a lateral pulsion (left-right). Each test is scored from 0 (if the animal has wholly lost function) to 4 (normal function). The sensorimotor function is considered normal when the score is 27–28 points, while a score of ≤15 indicates severe sensorimotor dysfunction [39].

### 2.7. Microdialysis and HPLC

A microdialysis probe consisting of an active region (polyacrylonitrile membrane, 40,000 Da pore size) of 3 mm was fabricated according to Maidment et al. (1989) [42]. On the day of microdialysis, the probe was inserted into the previously implanted guide cannula and perfused (2 µL/min) with fresh, sterile artificial cerebrospinal fluid (125 mM sodium chloride, 2.5 mM potassium chloride, 0.5 mM dibasic sodium phosphate, 5 mM monobasic sodium phosphate, 1 mM magnesium chloride, 0.2 mM ascorbic acid, 1.2 mM calcium chloride, pH 7.4). Two hours after probe implantation, dialysates were recovered every 30 min and according to each protocol (Figure 1).

Dialysates were processed to quantify glutamate concentration by HPLC. The dialysates were diluted with perchloric acid (2 N, 1:20). Then, 15 μL of the mixture was mixed with 10 μL of *o*-phthalaldehyde and *n*-acetylcysteine, agitated for 30 s, and injected into the solvent stream of an HPLC system. The HPLC system comprised a fluorescence detector operating at an excitation wavelength of 360 nm and an emission wavelength of 450 nm. The HPLC fluorometric detection procedure for amino-acid quantitation required the *o*-phthalaldehyde amino acids to be separated on a reversed-phase 3.9 mm × 150 mm column (Nova-Pack, 4 μm, C_18_, Waters^®^) using solution A (sodium acetate dissolved in milli-Q water, pH 5.05) as aqueous solvent, solution B (acetonitrile), and solution C (water milli-Q) as a gradient flow rate of 1 mL/min (Waters^®^ model 474) [43].

Glutamate concentrations for the short-term experiment were expressed as a percentage change from basal conditions obtained before induction of TBI or manipulation. Preliminary results indicated that severe TBI induces large fluctuations in glutamate release at different times. Therefore, we decided to analyze the glutamate release during the entire process of the microdialysis experiments of the short-term experiment (5 h) using the area under the curve (AUC). The AUC is an analysis that results from a trapezoidal formula (Graphpad Prism 8.0.1 Software, Dotmática, San Diego CA, USA) [44] and allows evaluating a phenomenon over time without outflow of the information contained in multiple measurements. Concerning the long-term experiment, the results obtained were expressed as extracellular glutamate concentrations (μM).

### 2.8. Nissl Histology

Under anesthesia (pentobarbital, 70 mg/kg i.p.), animals were perfused with 250 mL of saline (0.9%) and heparin solution (1 mg/L, Sigma-Aldrich, Cat # H3393, Mexico City, Mexico), followed by 250 mL of paraformaldehyde (4%, Sigma-Aldrich Cat # P6148) and glutaraldehyde (0.2%, Electron Microscopy Sci. Cat # 16210, Hatfield, PA, USA) in a phosphate buffer solution. After perfusion, the brain was dissected and kept in a paraformaldehyde solution at 4 °C. Serial 15 µm thick coronal sections were cut with a cryostat and processed to evaluate the cannula implantation site. Then, sections were immersed in Cresyl violet for 30 min. Subsequently, they were washed with distilled water to remove excess dye. Sections were dehydrated using increasing concentrations of alcohol (70%, 96%, and 100%) and 100% xylene (5 min in each solution). The samples were covered with synthetic resin and analyzed with an optical microscope.

### 2.9. Statistical Analysis

The statistical analysis was carried out by an investigator blinded to the experimental conditions. Data were expressed as the mean ± standard error. Mixed-effects analysis and one-way ANOVA were used to detect changes in glutamate concentration through the microdialysis short-term experiment. One-way ANOVA was used to identify significant differences in AUC and Neuroscore values. Two-way ANOVA was applied to detect significant differences in the body weight. Tukey’s test was performed to compare differences between groups, whereas Dunnett’s test was used to compare values with basal conditions. Fisher’s exact test was applied to identify significant changes in the mortality rate. a ROUT (Q = 1%) test was applied to detect outliers within the different experimental groups. A statistically significant difference was considered when *p* ≤ 0.05.

## 3. Results

Histological analysis showed that the microdialysis probes were correctly implanted in the cerebral cortex of the animals used in this study.

### 3.1. CBD Pretreatment Lessens TBI-Induced High Extracellular Glutamate Release in the Cortex Shortly after Trauma

Regarding in vivo microdialysis experiments, the SHAM group showed constant extracellular glutamate concentration during the 5 h of evaluation after manipulation (1.5 ± 0.16 µM, (F_(10,66)_ = 1.306, *p* = 0.2457) (Figure 2). The AUC for this experimental group was 426 ± 56 (Figure 3). The baseline values of all experimental groups were similar to the SHAM group (F_(4,30)_ = 0.5458, *p* = 0.7034). This finding supports that orally pretreatment of CBD did not modify the glutamate release under basal conditions.

Microdialysis experiments of the TBI group revealed a significant increase in glutamate release (738% ± 173%; *p* < 0.001 vs. basal values) in the dialysate collected 30 min after trauma. One outlier value at 30 min post-TBI was removed from the final analysis. Significant fluctuations were also detected at 3 and 4 h (514% ± 156%; *p* < 0.05 and 312% ± 132%, *p* = 0.993 vs. baseline, respectively) after TBI (F_(10,63)_ = 4.468, *p* < 0.0001, Figure 2). The AUC for this experimental group was 1348 ± 124 (*p* < 0.001 vs. SHAM) (F_(4,30)_ = 9.906, *p* < 0.0001, Figure 3).

The LFPI applied to the different experimental groups orally pretreated with CBD induced an increase in glutamate release 30 min after trauma. However, the over-release of this amino acid was lower when compared to TBI group (CBD50 + TBI, 314% ± 73%, *p* < 0.05; CBD100 + TBI, 222% ± 33%, *p* < 0.01; CBD200 + TBI, 345% ± 98%, *p* < 0.05) (F_(4,29)_ = 5.44, *p* = 0.0021, Figure 2). One outlier value from each experimental group at 30 min post-TBI and one value at 3 h post-TBI from the CBD200 + TBI group were removed from the final analysis. The global estimation of glutamate release throughout 5 h after trauma (AUC) revealed lower concentrations when compared to the TBI group (CBD50 + TBI, 1072 ± 114, *p* < 0.001; CBD100 + TBI, 574 ± 79, *p* < 0.001; CBD200 + TBI, 919 ± 181, *p* < 0.05) (F_(4,30)_ = 9.906, *p* < 0.0001, Figure 3).

According to the results obtained from this experiment, CBD at 100 mg/kg orally administered was considered the optimal treatment to be evaluated during the long-term experiment, as it was the most effective in preventing high post-TBI glutamate release.

### 3.2. CBD Pretreatment Does Not Modify the Short-Term TBI-Induced Changes in Body Weight and Sensorimotor Function, but Reduces Mortality

The animals from the SHAM group showed a body weight gain of 1.3% and a score of 26–28 points in the sensorimotor evaluation during the experimental procedure. Rats from the TBI group showed a 9.7% reduction in body weight and decreased sensorimotor function (13.57 ± 1.6 points, *p* < 0.001 vs. SHAM) (F_(4,30)_ = 14.82, *p* < 0.0001, Table 1) 2 days after TBI. This group showed a 50% of mortality rate post-TBI. As consequence of TBI, the different experimental groups orally pretreated with CBD (50, 100, or 200 mg/kg) presented a weight loss (6.1%, 7.7%, and 9.6%, respectively) and decreased sensorimotor score (16.4 ± 1.3 points, 14.7 ± 2.7 points, and 15.1 ± 2.2 points, respectively). These changes were not significantly different when compared to the TBI group (Table 1). The experimental groups pretreated with CBD showed a mortality rate lower when compared to the TBI group (CBD50 + TBI, 22%, *p* < 0.001; CBD100 + TBI, 22%, *p* < 0.001; CBD200 + TBI, 36%, *p* < 0.05).

### 3.3. CBD Administration Attenuates Long-Term High Glutamate Release after Severe TBI

Microdialysis on day 30 revealed a glutamate concentration of 0.82 ± 0.2 µM during the 3 h evaluation of the SHAM group. CBD + SHAM group presented similar glutamate extracellular levels (0.78 ± 0.15 µM, *p* > 0.999 vs. SHAM group) during microdialysis (F_(4,30)_ = 7.055, *p* = 0.0004). Concerning the TBI group, the glutamate concentration estimated 30 days after trauma (2 ± 0.3 µM) was 149% higher than the SHAM group (*p* < 0.01). In contrast, the glutamate concentration observed 30 days after severe TBI in rats receiving CBD was like the SHAM group (CBD + TBI, 0.73 ± 0.2 µM, *p* = 0.9983; CBD + TBI + CBD, 1 ± 0.2 µM, *p* = 0.944) (Figure 4).

### 3.4. CBD Administration Diminishes Long-Term Sensorimotor Deficit, Improves Body Weight Gain, and Reduces Mortality after Severe TBI

Rats in the SHAM group presented a body weight gain of 40.98% ± 4.3% on day 28 after manipulation (*p* < 0.001 vs. basal condition) and a Neuroscore of 27.3 ± 0.1 throughout the experiment. One animal in this group died during anesthesia. The CBD + SHAM group showed a body weight gain similar to the SHAM group (*p* = 0.752), and Neuroscore values were maintained at 26.7 ± 0.4 on day 28 after manipulation (Table 2).

Rats in the TBI group underwent a body weight loss (F_(4,30)_ = 14.82, *p* < 0.0001) registered 2 days post-TBI (−13.74% ± 1.9%, *p* < 0.001 vs. SHAM). Body weight increased (F_(4,30)_ = 6.956, *p* = 0.0004) progressively and achieved a gain of 21.61% ± 3.1% over initial body weight, albeit still lower than the SHAM group (*p* < 0.01). On day 2 post-TBI, the animals showed Neuroscore values of 11.29 ± 1.8 (*p* < 0.001 vs. SHAM group) that gradually increased and achieved 17.6 ± 1.1 points (*p* < 0.001 vs. SHAM group) on day 28 post-TBI (F_(4,30)_ = 28.43, *p* < 0.0001). This experimental group presented a mortality rate of 36% through the experimental procedure (Table 2).

Animals from the CBD + TBI and CBD + TBI + CBD groups showed changes in the body weight like TBI group. However, body weight augmentation at day 28 after trauma was higher (CBD + TBI, 39.38% ± 2.7%, *p* < 0.01; CBD + TBI + CBD, 27.31% ± 2.4%, *p* < 0.05 vs. TBI group) (F_(4,30)_ = 6.956, *p* = 0.0004). Rats from these experimental groups showed low sensorimotor capacity (CBD + TBI, 14.1 ± 1.1 points, *p* < 0.001; CBD + TBI + CBD, 14.57 ± 1.4 points, *p* < 0.001 vs. SHAM, respectively) on day 2 post-TBI. Neuroscore values gradually increased and, on day 28 post-TBI, were lower than the SHAM group but higher compared to the TBI group (*p* < 0.01 and *p* < 0.001, respectively) (F_(4,30)_ = 28.43, *p* < 0.0001). In contrast to the TBI group, rats receiving CBD administration showed a lower mortality rate (CBD + TBI, 22%, *p* < 0.01; CBD + TBI + CBD, 30%, *p* > 0.5) (Table 2).

## 4. Discussion

The present study supports that severe TBI induces an increase in extracellular glutamate concentration in the cortex in the short- and long-term after the trauma. Our microdialysis experiments revealed that the oral pretreatment with CBD attenuates the TBI-induced over-release of glutamate, an effect most effective at 100 mg/kg. These results support previous findings indicating that CBD pretreatment lessens glutamate release in synaptosomes obtained from the hippocampus of cocaine-treated rats [32].

Favorable results on the immediate consequences of TBI have been obtained with treatments with antioxidant molecules [45,46], anti-inflammatory drugs [47], and NMDA antagonists [48], as well as hypoxic preconditioning strategies [49]. Glutamate over-release is known to trigger neuroinflammation [50], oxidative stress [51], and neurodegeneration [52]. In turn, chronic excitotoxicity and neuroinflammation can lead to immuno-excitotoxicity [53]. Depending on the severity of the injury, TBI induces glutamate over-release that can cause excitotoxicity and significant neuronal death [54]. After trauma, long-term functional outcome (6–12 months) in patients correlates with early post-TBI excitotoxicity estimated by proton magnetic resonance spectroscopy [55]. Our microdialysis experiments showed an over-release of glutamate during the first minutes after severe TBI, an effect that remained evident 30 days after trauma. These results are consistent with previous studies in preclinical models [11,56] and patients with severe TBI [55,57]. Another study indicated that high potassium evoked glutamate release in the primary somatosensory cortex in male but not female rats at 28 days after TBI [58]. Future studies are necessary to investigate the influence of clinical conditions on the TBI-induced glutamate excitotoxicity.

The immediate increase in extracellular glutamate levels after severe TBI may be caused by the primary injury that favors the rupture of vessels, neurons, and BBB, facilitating the release of intracellular contents into the parenchyma [59,60]. On the other hand, persistent high glutamate release for several hours and days after TBI may result from dysregulation of glutamate reuptake in astrocytes by chronic dysfunction in astrocytic transporters (GLAST/GLT1) [61,62]. Indeed, experimental evidence supports the downregulation of GLUT-1 transporters 7 days post-TBI [63]. In addition, TBI induces high glutamate levels as a consequence of inactivating mitochondrial complex 2-oxoglutarate dehydrogenase (OGDHC) [64]. Dysregulation of intracellular calcium could also facilitate the progressive and sustained increase in glutamate concentrations [65].

Severe TBI results in sensorimotor dysfunction during the first days after trauma. Thereafter, patients showed gradual recovery of sensorimotor function, but more than 25% of patients persisted with some form of disability 2 years after trauma [66]. Preclinical studies support that animals submitted to a severe TBI still do not reach the conditions of the control group 30 days after trauma [40,67]. TBI also induces long-term changes in body weight gain that can be explained by hypermetabolism and catabolism [68] due to damage in hunger-satiety centers such as the lateral ventromedial hypothalamus. Furthermore, TBI induces a decreased expression of orexin/hypocretin [69], a hormone that regulates nutritional processes and short-term intake [70].

CBD is a multitarget molecule with different neuroprotective effects demonstrated in preclinical studies [71]. Our experiments revealed that the oral subchronic pretreatment with CBD at doses of 50 to 200 mg/kg reduced the TBI-induced glutamate release, an effect more evident at 100 mg/kg. Further experiments are essential to investigate if these effects are still evident at higher doses of CBD. Indeed, a previous study supports that CBD administration results in dual effects in the prepulse inhibition test (antipsychotic measure) in a mouse model of schizophrenia [72], an effect that depends on the doses applied. In addition, it is described that, at low concentrations, CBD is an agonist of 5-HT_1A_ receptors [73], facilitating inhibitory effects. In contrast, CBD at high concentrations acts as an inverse agonist of these receptors [74] and can induce excitatory effects.

CBD decreases glutamate concentration under in vitro hypoxia/ischemia conditions, an effect associated with activating adenosine A_1A_, CB_1_, and CB_2_ receptors [75]. Concerning the endocannabinoid system, studies indicate that CBD inhibits the enzymatic degradation and reuptake of anandamide [76]. Enhanced levels of anandamide can result in activation of CB_1_ receptors with a subsequent decrease of glutamate release [77,78]. Overall, an indirect anti-excitotoxic mechanism of CBD through increased levels of endogenous endocannabinoids cannot be ruled out.

Calcium plays a vital role in releasing neurotransmitters, including glutamate [79]. It is known that the exposure to CBD decreases calcium release from the endoplasmic reticulum [80]. On the other hand, NCX2-3 transporters play an essential role in the calcium–sodium exchange and neuronal excitability [81], and the inhibition or downregulation of this transporter augments intracellular calcium, favoring neuronal death [82,83]. A study supports that the intracerebroventricular pretreatment with CBD increases the expression of the NCX2 transporter in the cerebral cortex and facilitates the preservation of BBB in animals subjected to hypoxia-ischemia [84]. Then, a decrease in calcio release induced by CBD can explain the lower over-release of glutamate in animals with TBI.

Administration of CBD in experimental models of hypoxic/ischemic events results in long-term anti-inflammatory, anti-excitotoxic, and neuroprotective effects associated with improved coordination, memory, and sensorimotor activity [25,85]. We found that rats orally administered with CBD at 100 mg/kg daily for 7 days before TBI showed extracellular glutamate levels like the control group when evaluated long-term after trauma. This effect was associated with improved sensorimotor performance and body weight gain, as well as lower mortality. It is essential to note that the CBD dose of 100 mg/kg orally applied daily for 7 days in the control group did not induce changes in the different variables evaluated, supporting that this treatment did not cause side-effects under these experimental conditions. This finding is consistent with the results obtained in previous studies administering oral CBD chronically (15 or 30 mg/kg for 34 days) [86].

The decreased TBI-induced acute over-release of glutamate in animals pretreated with CBD can result in lower excitotoxicity and long-term protective effects after trauma [87,88]. The observed long-term protective effects after TBI may be explained since CBD induces anti-inflammatory effects mediated by low expression of TNFα, NF-κB IL-1β, GFAP, and AQP4 and an improved BBB viability [31,89]. In addition, CBD reduces the expression of proapoptotic proteins BCL-2/Bax, and caspase-3, increasing the activity of antioxidant enzymes such as superoxide dismutase and catalase [90]. CBD also decreases mitochondrial dysfunction associated with neuroinflammation [91]. Another possibility to explain the long-term protective effect found in the present study is that CBD avoids the TBI-induced dysregulation of glutamate uptake in astrocytes [61,62,63]. Overall, it is possible to indicate that CBD-induced effects result in long-term functional improvement after trauma.

Our results support that oral pretreatment with CBD facilitates the recovery of body weight gain and sensorimotor function long-term after severe TBI. In this regard, chronic CBD treatment increased body weight gain in a rat model of depression [92]. Similarly, chronic CBD administration improved weight gain, induced anxiolytic effects, decreased hyperglycemia, and increased plasma insulin in a diabetes model [93]. On the other hand, acute administration of CBD via two different routes (injury site and systemic) improved long-term learning, memory, and vestibulo-motor functions after TBI. These effects were associated with diminished lesion volume and neuronal loss [36]. Therefore, it is possible that the effects of CBD on body weight gain and sensorimotor functions post-TBI are associated with increased cell viability and neuroprotection. An important finding from the long-term experiment is that CBD pretreatment (CBD + TBI group) was more effective than the pre/post-TBI treatment scheme (CBD + TBI + CBD group). Similar effects were found when CBD was applied as pretreatment, but not pre- and post-treatment in an experimental model of seizures [94]. These results lead to suggest that the protective effects of CBD are not evident once the insult is established, probably because of changes in transduction mechanisms. Further experiments are required to corroborate this idea.

We found that CBD pretreatment reduced the TBI-induced mortality. Previous studies support that CBD decreases mortality in experimental models of sepsis [95], Dravet syndrome, and seizures [96]. Severe TBI is associated with a high mortality rate as a consequence of multiple factors such as increased intracranial pressure, edema formation, damage to the BBB, and hypoxia/ischemia [97]. Hypoxia favors high glutamate levels [98] and axonal damage. Post-TBI diffuse axonal damage in brainstem [99] must be associated with respiratory depression and cardiovascular changes that induce mortality. Another condition of high mortality in patients with severe TBI is the post-trauma arterial hypertension [100]. It is described that CBD injection into the bed nucleus of the stria terminalis restores the baroreflex function through its action on 5HT_1A_ receptors [101]. In addition, CBD induces a positive effect on hemodynamic variables in an experimental model of hypoxia/ischemia [102]. This group of evidence suggests that CBD induces beneficial effects on systemic circulation and reduces the TBI-induced mortality. However, further studies are essential to support that oral pretreatment with CBD is a good therapeutic strategy to reduce the cardiovascular impairments and mortality after a severe TBI.

TBI induces dysautonomia and systemic inflammation that facilitate gastrointestinal events such as dysmotility and increased mucosal permeability [103]. It is suggested that microbiota plays a relevant role in the TBI-induced dysfunction in the brain–gut axis [104]. Indeed, preclinical studies support that the risk to develop epilepsy as a consequence of a severe TBI is associated with the preexistent gut microbiome profile [105]. On the other hand, CBD decreases central and peripheral inflammation, an effect also detected in gastrointestinal tract without changes in the gut microbiota [106]. Future studies focusing on supporting the beneficial effects of orally administered CBD must consider the gut microbiome profile.

It is important to notice that the oil used in this study had a low concentration of CBDV (0.65%). This is not unexpected since CBDV has been detected by HPLC-UV in different hemp oils [107]. CBDV induces anticonvulsant effects without impairing motor function at 50–200 mg/kg in experimental models of seizures [108]. Studies support that oral administration of CBDV at 400 mg/kg in rats induces anticonvulsant effects in pentylenetetrazol-induced seizures [109]. In the present study, we orally applied 0.65 mg/kg of CBDV in combination of 100 mg/kg of CBD. CBDV presents a low absorption and an important interaction with metabolic enzymes that result in low bioavailability [110]. According to its pharmacokinetics, we expect that CBDV is not implied in the CBD-induced effects found in the present study. Nevertheless, studies must be carried out to support this notion.

## 5. Conclusions

CBD administration decreases short- and long-term glutamate over-release after severe TBI, an effect associated with improvement of sensorimotor activity and body weight gain, as well as lower mortality. Future studies are essential to elucidate the mechanisms via which CBD exerts these effects and whether other mechanisms, such as antiinflammation, antioxidative stress, and neuroprotection, are involved. Our study suggests a neuroprotective effect of CBD short- and long-term after a severe TBI. These findings support orally administered CBD as a therapeutic strategy to prevent long-term consequences after TBI in the high-risk population, such as military personnel and contact sport athletes. It will also be interesting to determine if CBD neuroprotection is sufficient to prevent the development of long-term disorders following TBI, such as Alzheimer’s disease, Parkinson’s disease, and post-traumatic epilepsy.

## Figures and Tables

**Figure 1 pharmaceutics-14-01609-f001:**
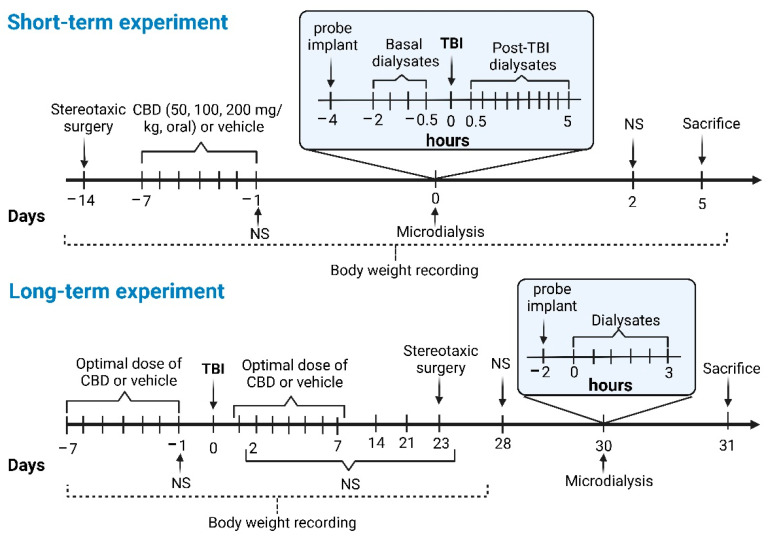
Timeline of the short- and long-term experimental design to determine the effects of oral administration of CBD on severe traumatic brain injury (TBI). Throughout the protocols, the body weight and sensorimotor function of the animals were registered. For the short-term experiment, a guide cannula was implanted in the cerebral cortex. Seven days after surgery, CBD or vehicle was administered every 24 h for 7 days. Twenty-four hours after the last CBD administration, a microdialysis experiment and TBI induction were performed. Five days after TBI, animals were sacrificed and perfused for histological analyses. For the long-term experiment, rats received the optimal dose of CBD or vehicle every 24 h for 7 days. TBI was induced 24 h after the last administration. One group received CBD treatment for an additional 7 days after TBI. On day 23 post-TBI, a guide cannula was implanted in the cerebral cortex, and microdialysis was performed 1 week later. Animals were sacrificed and perfused on day 31 post-TBI for histological analysis. The figure was created using BioRender.com.

**Figure 2 pharmaceutics-14-01609-f002:**
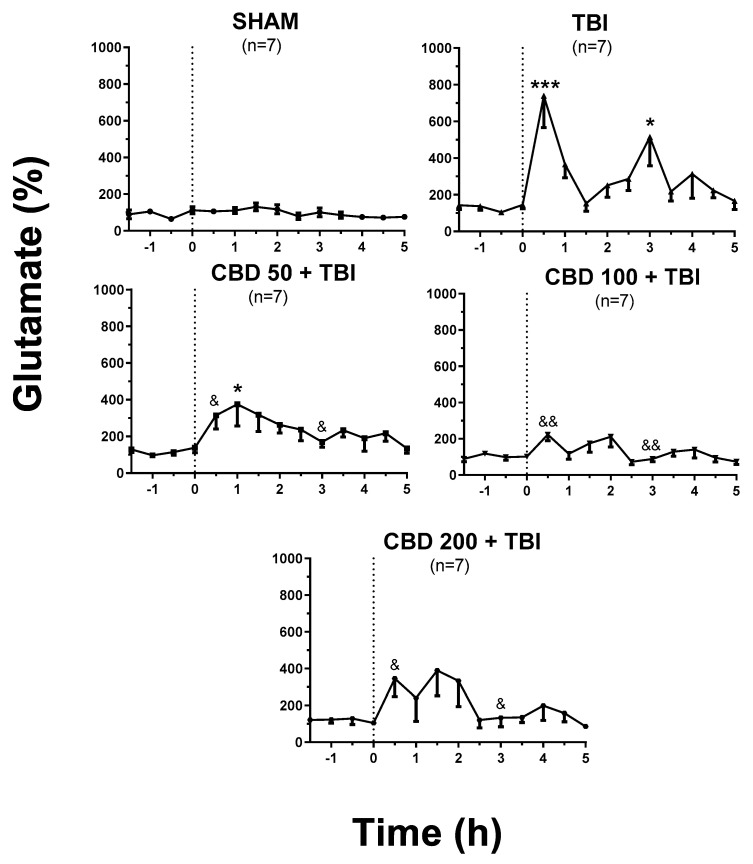
Effects of oral administration of CBD on glutamate over-release induced by severe TBI. The SHAM group maintained a constant glutamate release throughout the microdialysis experiment. The TBI group showed an increase in glutamate release after the induction of severe TBI and subsequent fluctuations throughout the experimental period. The groups pretreated with CBD showed an enhanced glutamate release after TBI. However, the changes were lower when compared with the TBI group. This effect was more evident in animals orally pretreated with 100 mg/kg of CBD. Results were analyzed with mixed-effects analysis (F_(52,385)_ = 2.039) and one-way ANOVA. They are presented as the mean ± standard error of the percent of change from basal conditions; * *p* < 0.05 vs. basal condition (F_(10,65)_ = 1.609, *p* = 0.123); *** *p* < 0.001 vs. basal condition (F_(10,63)_ = 4.468, *p* < 0.0001). At 30 min after TBI, ^&^
*p* <0.05, ^&&^
*p* <0.01 vs. TBI group (F_(4,26)_ = 6.61, *p* = 0.0008). At 3 h after TBI, ^&^
*p* < 0.05, ^&&^
*p* < 0.01 vs. TBI group (F_(4,29)_ = 5.44, *p* = 0.0021). CBD, cannabidiol; TBI, traumatic brain injury.

**Figure 3 pharmaceutics-14-01609-f003:**
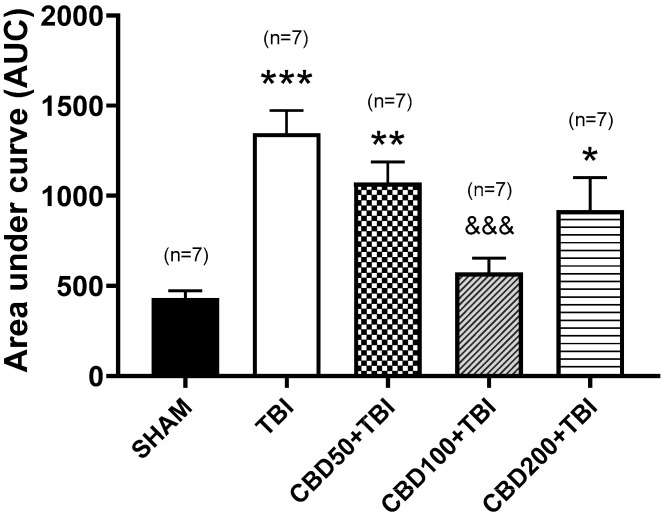
Representation of the glutamate release during the entire process of the microdialysis experiments of the short-term experiment through the area under curve (AUC). The TBI group showed an increased overall glutamate release. Experimental groups pretreated with CBD showed lower AUC values when compared with the TBI group. This effect was more evident in the group pretreated with 100 mg/kg. * *p* < 0.05 vs. SHAM group, ** *p* < 0.01 vs. SHAM group, *** *p* < 0.01 vs. SHAM group; ^&&&^ *p* < 0.001 vs. TBI group (F_(4,30)_ = 9.906, *p* < 0.0001, one-way ANOVA). CBD, cannabidiol; TBI, traumatic brain injury.

**Figure 4 pharmaceutics-14-01609-f004:**
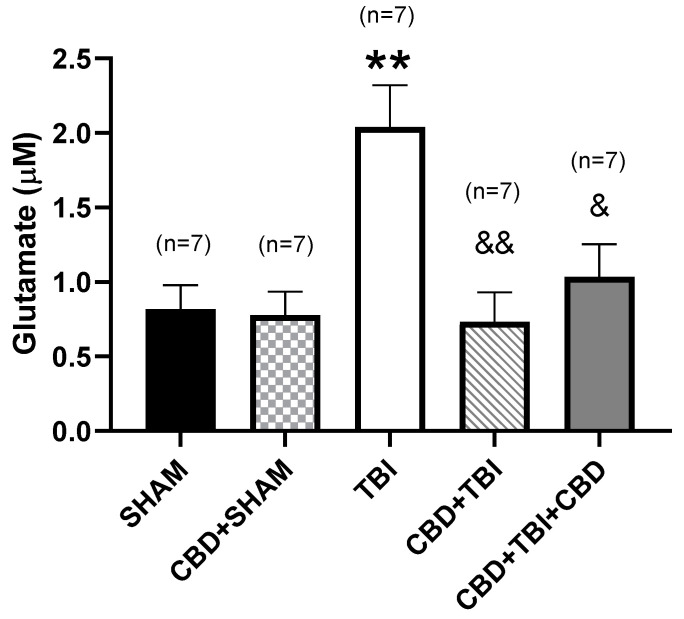
Effects of CBD administration on glutamate concentration at day 30 post-TBI. The SHAM and CBD + SHAM groups showed a similar glutamate concentration. TBI group showed a higher glutamate concentration when compared to the SHAM group. In contrast, CBD + TBI and CBD + TBI + CBD groups showed a glutamate concentration similar to the SHAM group. Values represent the mean ± standard error. ** *p* < 0.01 vs. SHAM; ^&^ *p* < 0.05 vs. TBI, ^&&^ *p* < 0.01 vs. TBI (F_(4,30)_ = 7.055, *p* = 0.0004, one-way ANOVA). CBD, cannabidiol; TBI, traumatic brain injury.

**Table 1 pharmaceutics-14-01609-t001:** Effects of oral administration of CBD on severe TBI-induced changes in body weight, sensorimotor function, and mortality rate.

Groups (at the Beginning of the Experiment)	Body Weight in g (% vs. Basal) (*n* = 7)	Neuroscore (*n* = 7)	Percentage Mortality (after TBI)
**SHAM** (*n* = 7)	294 ± 6.7 (+1.3%)	27.0 ± 0.3	0%
**TBI** (*n* = 14)	263 ± 10.5 *** (−9.7%) ^@@@^	13.5 ± 1.6 ^###^	50%
**CBD50 + TBI** (*n* = 9)	269 ± 11.8 ** (−6.1%) ^@^	16.4 ± 1.4 ^##^	22.2% ^&&&^
**CBD100 + TBI** (*n* = 9)	253 ± 11.5 *** (−7.2%) ^@@^	14.8 ± 2.8 ^###^	22.2% ^&&&^
**CBD200 + TBI** (*n* = 11)	268 ± 23.4 *** (−9.6%) ^@@@^	15.1 ± 2.3 ^###^	36.3% ^&^

Values represent the mean ± standard error. ** *p* < 0.01, *** *p* < 0.001 vs. SHAM (F_(4,30)_ = 14.82, *p* < 0.0001, one-way ANOVA); ^@^ *p* < 0.5, ^@@^ *p* < 0.01, ^@@@^ *p* < 0.001 vs. basal values (F_(4,30)_ = 14.82, *p* < 0.0001, two-way ANOVA); ^##^ *p* < 0.01, ^###^ *p* < 0.001 vs. SHAM group (F_(4,30)_ = 14.82, *p* < 0.0001, one way-ANOVA); ^&^ *p* < 0.05, ^&&&^ *p* < 0.001 vs. TBI group (Fisher’s exact test).

**Table 2 pharmaceutics-14-01609-t002:** Effect of pre- or pre-post CBD treatment on severe TBI-induced changes in body weight, sensorimotor function, and mortality at day 28 after trauma.

Groups (at the Beginning of the Experiment)	Body Weight in g (% vs. Basal) (*n* = 7)	Neuroscore (*n* = 7)	Percentage Mortality (after TBI)
**SHAM** (*n* = 8)	401 ± 13.5 (+40.9%) ^@@@^	26.8 ± 0.4	0%
**CBD + SHAM** (*n* = 7)	409 ± 18 (+35.8%) ^@@@^	26.7 ± 0.4	0%
**TBI** (*n* = 11)	329 ± 15 ^##^ (+21.6%) ^@@@^	17.5 ± 1.1 ***	36.3%
**CBD + TBI** (*n* = 9)	375 ± 15 ^&&^ (+39.3%) ^@@@^	22.0 ± 0.6 ***^/ə^	22.2% ^●●^
**CBD + TBI + CBD** (*n* = 10)	398 ± 24 ^&^ (+27.3%) ^@@@^	22.4 ± 0.7 **^/ə^	30%

Values represent the mean ± standard error. ^##^ *p* < 0.01 vs. SHAM; ^&^ *p* < 0.05, ^&&^ *p* < 0.01 vs. TBI group (F_(4,30)_ = 6.956, *p* = 0.0004, one-way ANOVA); ^@@@^ *p* < 0.001 vs. basal values (F_(4,30)_ = 6.848, *p* = 0.0005, two-way ANOVA); ** *p* <0.01, *** *p* < 0.001 vs. SHAM; ^ə^ *p* < 0.05 vs. TBI group (F_(4,30)_ = 28.43, *p* < 0.0001, one-way ANOVA); ^●●^ *p* < 0.01 (Fisher’s exact test).

## Data Availability

Not applicable.

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
