# Peer review of "Cannabidiol Reduces Short- and Long-Term High Glutamate Release after Severe Traumatic Brain Injury and Improves Functional Recovery"

_pharmaceutics, 2022, doi:10.3390/pharmaceutics14081609_

Round 1

Reviewer 1 Report

Dear authors,

After the review report, I have several comments: you should include numerical data in the abstract; you did not make any mention about the bioavailability process in the introduction/discussion; you should include comments about the bioavailability process and presents additional data (microbiota bioactivity, e.g.) as a future paper valorization; you should include references in all Materials and Methods sections; how was realized figure 1 - copyright is necessary?!. Best regards!

Author Response

You should include numerical data in the abstract

Response: Numerical data were included in the abstract.

You did not make any mention about the bioavailability process in the introduction/discussion. You should include comments about the bioavailability process and presents additional data (microbiota bioactivity, e.g.) as a future paper valorization.

Response: The new version of the manuscript includes information about the bioavailability of CBD when orally administered. In addition, the discussion section now includes a paragraph indicating the relevance of microbiota in future studies focused to support the beneficial effect of orally administered CBD.

You should include references in all Materials and Methods sections.

Response: The new version of the manuscript includes references through the Materials and Methods section.

How was realized figure 1 - copyright is necessary?

Response: The legend of figure 1 now indicates that Figure 1 was created with BioRender.com. Then, copyright is not necessary.

Reviewer 2 Report

Comments to the Authors 

Re: Cannabidiol reduces short- and long-term high glutamate release after traumatic brain injury and improves functional recovery

Cindy Santiago-Castaneda et al. examined the effects of CBD pretreatment and post administration at several doses (50-200 mg/kg) on extracellular glutamate levels by microdialysis method as well as sensorimotor dysfunction following severe traumatic brain injury (TBI). They report a dose specific effect of CBD at the 100 mg/kg dose, with lower or no effect at the other doses 50 or 200 mg/ kg CBD. However vigorous statistics were not done to make that claim as decreases in glutamate levels were also observed at the other doses in short and long-term experiments. They also observed relatively no differences between the pretreated CBD group at 100 mg/kg with the pre and post-treated CBD group which is not discussed.

Overall the results look convincing and the English is well written but the authors could be more concise, less redundant, and the organization and presentation of the results are haphazard.

The discussion reiterates important points that belong in the introduction and lacks explanation of their results. For example, why is there a dose dependent effect of CBD pretreatment? Is there a proposed effect on endocannabinoid levels which may retroactively shut down glutamate release?  Also what is the source of the CBD? Is it pure or is it an extract from leaves containing other phytocananbinoids? Was there more neuroprotection in the CBD 100 mg/kg group over the other CBD treated groups? Is there a correlation between glutamate concentration and amount of injury or dose vs injury? A two-way ANOVA would address this issue as there are two factors: Drug dose and treatment type (pre and post). This is not clear in the results. Highlights on the pdf points to main issues to assist the authors. After a thoughtful rewrite of the discussion, and statistics in place, the paper needs to be re-reviewed.

Specific Comments.

Abstract.

First, please reorganize the results starting with the short-term experiment then the long-term experiment as presented in the methods and results followed by the glutamate levels.

Line 21. The figure 3 shows a reduction in peak levels at 30 min at all doses of CBD and higher reduction at 3 and 5 h by the 100 mg dose. I see lower peaks at these times also at 50 and 200 mg/kg doses that do not appear to be significantly higher then the lower levels achieved with CBD 100 mg/kg dose (100 vs 200?). Please revise as a decrease was observed at all doses tested. You would have to run a two way ANOVA to determine whether the dose of time is significant which was not done.

Lines 22- 24. Pretreatment with 50 and 200 also showed reduction but not as well as the CBD 100 group. Just say something like: There was a reduction in glutamate concentration levels at all CBD doses tested but with higher potency at when 100 mg/kg was administered. Please revise the presentation of the result in the abstract and in the results section to compare the actual two means followed by the p value. Not 149% vs. sham. Its 149% compared to baseline. What is the baseline? It’s not p<0.01 vs. TBI. You need to put in the numbers or just state in the abstract that the reduction was significant and save all the values for the text in the results sections.  All groups suffered from sensorimotor deficits at 48 h and all groups show recovery but highest with CBD pretreatment whether they had it after TBI or not.

You mention that CBD+TBI and CBD+TBI+CBD were similar suggesting the pre-treatment was sufficient to lessen the adverse sequelae of TBI. This should be added to the conclusion.

Methods:

This section needs some reorganization.

2.1 Animals

2.2 Stereotaxic surgery

2.3 Induction of severe TBI

2.4 Experimental groups (All treatment groups should be here in one paragraph not bouncing back and forth. (what does p.o. mean? Is it intraperitoneal? (i.p.). with the Experimental Design with the two schematic figures.

2.5 Microdialysis

2.6 Histology

2.7 Statistics - Two way ANOVA is the correct test but it is not clear how it was conducted and for which factors. Each needs to be identified and the F values with degrees of freedom must be included under each result in the text and legends.

Results.

Presentation of the results is haphazard. Please present the data under each topic line and not mix it up. 3.1 states pretreatment lessens glutamate. OK but the first sentence is about the weight then the figure for the weight and it’s description appears below as Figure 5!!

If you want to mention the weight loss and recovery then show it as Figure 1. The section should be titled Weight.

Then Neuroscore as 3.2 and the data as Fig. 3 or talk about it after the glutamate levels under its own topic.

Then 3.3 The glutamate concentration levels first short term then long term

Figure 3 is convincing which shows decreases under all doses with better efficacy for 100 dose. what does glutamate % stand for? % of what? Where is the F data? You cannot present p levels without the data! Give us the number (Means and SEMs) 149 ± %? You must do this for each experiment and figure.

Fig. 6. On the neuro-scoring, there is no difference between CBD+TBI and CBD+TBI+CBD treatment groups suggesting the initial pretreatment was responsible for the protection? The same observation is shown in Figs 3, 4  and 7.  Also Figure 4 appears to contradict Figure 3 where clearly 50 mg/kg reduces glutamate at 3 h compared to TBI so the 2nd peak is attenuated by all doses and the 3rd peak is lowered at the 100 dose. Please revise. Also please enter the statistical test and N for each group in each figure legend.

Histology. Only the placement is mentioned. What happened with the histology? Was there less injury with pre and post-treatment?

Discussion.

Some of the discussion is redundant with the introduction and some belongs in the introduction to make room for discussion of the results. For example, Combine lines 51-52 with Discussion line 362 and up to 368. The next part talks about the microdialysis results - good.

The first paragraph should summarize the main findings which is not about the weight (which was introduced as a non-sequetor) and does not even need to be mentioned in the discussion. First sentence OK next two delete then go to the microarray results.

Lines 397-401 move up to first paragraph - you are bouncing back and forth again.

Lines 402 – 411 should move down after you finish talking about the doses. Try to give an explanation as to why you think there was a dose effect which you did not prove statistically.

I have highlighted points in the discussion that needs much better organization that flows with the results in the same order that it was presented. After a thoughtful rewrite of the discussion, and statistics in place the paper needs to be re-reviewed.

Author Response

However vigorous statistics were not done to make that claim as decreases in glutamate levels were also observed at the other doses in short and long-term experiments. They also observed relatively no differences between the pretreated CBD group at 100 mg/kg with the pre- and post-treated CBD group which is not discussed.

Response: Thanks for the comment. We reorganize the statistical analysis of the results. They analysis revealed significant changes in the different parameters evaluated in short- and long-term experiments. We also discussed that CBD pretreatment was more effective than pre- and post-treatment.

Overall the results look convincing and the English is well written but the authors could be more concise, less redundant, and the organization and presentation of the results are haphazard.

Response: We reorganize the manuscript according to the reviewer´s comment.

-The discussion reiterates important points that belong in the introduction and lacks explanation of their results. For example, why is there a dose dependent effect of CBD pretreatment?

Response: The new version of the manuscript includes an explanation about the dose-dependent effects of CBD on glutamate release at 50 and 100 mg/kg. In addition, we included an explanation of the lack of effect at 200 mg/kg.

-Is there a proposed effect on endocannabinoid levels which may retroactively shut down glutamate release?  

Response: The discussion section now includes a paragraph describing the effect of CBD on the endocannabinoid system that may explain the reduced concentrations of glutamate.

-Also what is the source of the CBD? Is it pure or is it an extract from leaves containing other phytocananbinoids?

Response: The new version of the manuscript indicates that CBD was donated by HempMeds. It is a purified hemp oil extract containing 99.35% CBD and 0.65% Cannabidivarin (CBDV). In the discussion section, we discussed the relevance of applying CBD with CBDV.

-Was there more neuroprotection in the CBD 100 mg/kg group over the other CBD treated groups? Is there a correlation between glutamate concentration and amount of injury or dose vs injury? A two-way ANOVA would address this issue as there are two factors: Drug dose and treatment type (pre and post). This is not clear in the results. Highlights on the pdf points to main issues to assist the authors. After a thoughtful rewrite of the discussion, and statistics in place, the paper needs to be re-reviewed.

Response: We appreciate the reviewer´s comment. For the short-term experiment we did not correlate the TBI-induced glutamate release with the alterations induced by TBI (sensorimotor dysfunction, body-weight reduction, and mortality). We consider that the temporal difference between the TBI-induced changes in glutamate release (measured on the day of TBI induction) and the sensorimotor dysfunction, body-weight and mortality (estimated two days after TBI induction) could represent a bias in the interpretation of the data.

For the long-term experiment, we tested the hypothesis that CBD dose influences the reduction of glutamate release using one-way ANOVA. There are two factors for this experiment: dose and type of treatment (pre- and pre-post). However, the pre- and post-treated groups have only one level in the dose factor (100 mg/kg). The two-way ANOVA would test whether dose, type of treatment, or their interaction influences glutamate concentration/sensorimotor function. In the present study, the question we can answer is whether the 100 mg/kg dose is better when administered as pre- or pre-post-treatment, and for this reason, we consider a one-way ANOVA appropriate, considering the type of treatment as a factor.

Specific Comments.

Abstract.

-First, please reorganize the results starting with the short-term experiment then the long-term experiment as presented in the methods and results followed by the glutamate levels.

Response: The abstract was rewritten according to the reviewer´s comment.

Line 21. The figure 3 shows a reduction in peak levels at 30 min at all doses of CBD and higher reduction at 3 and 5 h by the 100 mg dose. I see lower peaks at these times also at 50 and 200 mg/kg doses that do not appear to be significantly higher then the lower levels achieved with CBD 100 mg/kg dose (100 vs 200?). Please revise as a decrease was observed at all doses tested. You would have to run a two way ANOVA to determine whether the dose of time is significant which was not done.

Response: Thanks for the comment. We did a two-way ANOVA as the reviewer suggested. ROUT (Q=1%) test was applied to detect outliers within the different experimental groups. The outlier values identified with this analysis were removed and the final statistical analysis revealed significant differences between the groups receiving CBD and TBI alone. We agree with the reviewer that glutamate release showed fluctuations through the microdialysis experiment of the short-term effects. Therefore, we decided to analyze the glutamate release during the entire process of the microdialysis experiments of the short-term experiment (5 h) using the area under the curve (AUC). The AUC is an analysis that results from a trapezoidal formula (Graphpad Prism 8.0.1 Software) and allows to evaluate a phenomenon over time without outflow the information contained in multiple measurements. Concerning the long-term experiment, the results obtained were expressed as extracellular glutamate concentrations (μM). This information is included in the new version of the manuscript.

Lines 22- 24. Pretreatment with 50 and 200 also showed reduction but not as well as the CBD 100 group. Just say something like: There was a reduction in glutamate concentration levels at all CBD doses tested but with higher potency at when 100 mg/kg was administered.

Response: The idea suggested by the reviewer was included in the abstract.

Please revise the presentation of the result in the abstract and in the results section to compare the actual two means followed by the p value. Not 149% vs. sham. Its 149% compared to baseline. What is the baseline? It’s not p<0.01 vs. TBI. You need to put in the numbers or just state in the abstract that the reduction was significant and save all the values for the text in the results sections.  

Response: In the short-term experiment, the results were compared with the baseline values because glutamate release was estimated before (baseline) and after TBI. In the long-term experiment, the animals were submitted to a TBI. On day 23 post-TBI, a guide cannula was implanted in the cerebral cortex, and microdialysis was performed one week later. According to this experimental protocol, baseline values were not obtained for the long-term experiment. The results obtained from the long-term experiment were compared with those obtained from the sham group, i.e., animals manipulated as described for the TBI group, except that TBI was not induced.

All groups suffered from sensorimotor deficits at 48 h and all groups show recovery but highest with CBD pretreatment whether they had it after TBI or not. You mention that CBD+TBI and CBD+TBI+CBD were similar suggesting the pre-treatment was sufficient to lessen the adverse sequelae of TBI. This should be added to the conclusion.

Response: We rewrote the abstract in order to indicate the changes induced by the pre- and pre-post-treatment of CBD in the long-term experiment. We included the following sentence: “Indeed, pre-treatment with CBD was sufficient to lessen the adverse sequelae of TBI”.

Methods:

This section needs some reorganization.

2.4 Experimental groups (All treatment groups should be here in one paragraph not bouncing back and forth. (what does p.o. mean? Is it intraperitoneal? (i.p.). with the Experimental Design with the two schematic figures.

Response: The section of Methods was reorganized as suggested by the reviewer. Concerning the question, “What does p.o. mean?”, through the first version of the manuscript we used the abbreviation p.o. from the Latin term “Per os” that means "through the mouth". In Pharmacology, this abbreviation is very common. In the new version of the manuscript, we changed p.o. by “oral administration” in order to avoid confusion.

Concerning the description of the experimental groups, we described them in two consecutive sections: 2.4. Experimental groups for the short-term experiment; 2.5 Experimental groups for the long-term experiment. We consider that all the experimental groups cannot be in one paragraph because they were designed for different purposes and the experimental conditions are different. To clarify this situation, we included a rationale for each experiment before the description of the experimental groups.

2.7 Statistics - Two way ANOVA is the correct test but it is not clear how it was conducted and for which factors. Each needs to be identified and the F values with degrees of freedom must be included under each result in the text and legends.

Response: The Materials section now includes a description of all the statistical analyses used for the present study. F vales with degrees of freedom are now indicated for each result in the text.

Results.

Presentation of the results is haphazard. Please present the data under each topic line and not mix it up. 3.1 states pretreatment lessens glutamate. OK but the first sentence is about the weight then the figure for the weight and it’s description appears below as Figure 5!!

If you want to mention the weight loss and recovery then show it as Figure 1. The section should be titled Weight.

Then Neuroscore as 3.2 and the data as Fig. 3 or talk about it after the glutamate levels under its own topic.

Then 3.3 The glutamate concentration levels first short term then long term

Response: We appreciate the reviewer´s comment. The Results section was rewritten in order to clarify the information obtained. We describe the results obtained according to the experiment: 3.1 and 3.2 for short-term, 3.3 and 3.4 for long-term. We consider this sequence because the experiments were designed for different purposes and the conditions were different. The Results section was rewritten as follows:

3.1. CBD pretreatment lessens TBI-induced high extracellular glutamate release in the cortex short-term after trauma

3.2. CBD pretreatment does not modify the short-term TBI-induced changes in body weight and sensorimotor function, but reduces mortality.

3.3. CBD administration attenuates long-term high glutamate release after severe TBI

3.4. CBD administration diminishes long-term sensorimotor deficit, improves body weight gain and reduces mortality after severe TBI

Figure 3 is convincing which shows decreases under all doses with better efficacy for 100 dose. what does glutamate % stand for? % of what? Where is the F data? You cannot present p levels without the data! Give us the number (Means and SEMs) 149 ± %? You must do this for each experiment and figure.

Response: In the section of methods (see 2.7 Microdialysis and HPLC), it is indicated that glutamate concentrations for the short‐term experiment were expressed as a percentage change from basal conditions obtained before induction of TBI or manipulation.

We included the F data, means and SEM, etc. according to the reviewer´s comment.

Fig. 6. On the neuro-scoring, there is no difference between CBD+TBI and CBD+TBI+CBD treatment groups suggesting the initial pretreatment was responsible for the protection? The same observation is shown in Figs 3, 4  and 7.  Also Figure 4 appears to contradict Figure 3 where clearly 50 mg/kg reduces glutamate at 3 h compared to TBI so the 2nd peak is attenuated by all doses and the 3rd peak is lowered at the 100 dose. Please revise. Also please enter the statistical test and N for each group in each figure legend.

Response: We agree with the reviewer that, according to the results obtained, it is possible to suggest the initial pretreatment was responsible for the protection. This finding is explained in the discussion section.

We re-analyze the results obtained. This new analysis supports the reviewer´s observations for the short-term experiment.

We included the statistical test and N per group in each figure.

Histology. Only the placement is mentioned. What happened with the histology? Was there less injury with pre- and post-treatment?

Response: The main goal of the present study was to investigate the effects of CBD in high glutamate release, sensorimotor dysfunction, body-weight alterations, and mortality induced by severe TBI. The histology was carried out with the purpose to identify the place of cannula implantation. We did not include a further evaluation of the brain injury induced by TBI. We have in process a study focused to determine the effects of CBD in neuroinflammation and neuronal damage.

Discussion.

Some of the discussion is redundant with the introduction and some belongs in the introduction to make room for discussion of the results. For example, Combine lines 51-52 with Discussion line 362 and up to 368. The next part talks about the microdialysis results - good.

Response: The following sentence was moved from the introduction to the discussion section, according to the reviewer´s comment: “Favorable results on the immediate consequences of TBI have been obtained with treatments with antioxidant molecules, anti-inflammatory drugs, and NMDA antagonists, as well as hypoxic preconditioning strategies.”

The first paragraph should summarize the main findings which is not about the weight (which was introduced as a non-sequetor) and does not even need to be mentioned in the discussion. First sentence OK next two delete then go to the microarray results. Lines 397-401 move up to first paragraph - you are bouncing back and forth again.

Response: The first paragraph of discussion was modified as follows: “The results in the present study support that severe TBI induces an increase in extracellular glutamate concentration in the cortex in the short- and long-term after the trauma. Our microdialysis experiments confirmed that the oral pretreatment with CBD 100 attenuates the short-term induced increase in glutamate release after TBI, an effect most effective at mg/kg. These results support previous findings indicating that CBD pretreatment lessens glutamate release in synaptosomes obtained from the hippocampus of cocaine-treated rats [30].

Lines 402 – 411 should move down after you finish talking about the doses. Try to give an explanation as to why you think there was a dose effect which you did not prove statistically.

I have highlighted points in the discussion that needs much better organization that flows with the results in the same order that it was presented. After a thoughtful rewrite of the discussion, and statistics in place the paper needs to be re-reviewed.

Response: The discussion was rewritten according to the reviewer´s comment. We really appreciate this comment.

Reviewer 3 Report

In the present manuscript, the main result that the authors show is that CBD administration decreases short‐ and long‐term glutamate over‐release after severe TBI. It is a very novel result and it is very well thought out.

Since the main result is based on glutamate values, in addition to the methodological reference a brief description of the methodology used to quantify glutamate should be described in the methodology section (chromatographic conditions, type of detector, treatment of dialyzed samples).

In results, during the microdialysis in short term experiment, it is not clear if the baseline values ​​of animals treated with CBD (either 50, 100 or 200) before TBI were similar to the SHAM group

Author Response

In the present manuscript, the main result that the authors show is that CBD administration decreases short‐ and long‐term glutamate over‐release after severe TBI. It is a very novel result, and it is very well thought out.

Response: We appreciate the reviewer´s comment.

Since the main result is based on glutamate values, in addition to the methodological reference a brief description of the methodology used to quantify glutamate should be described in the methodology section (chromatographic conditions, type of detector, treatment of dialyzed samples).

Response: Description of the procedure used to quantify glutamate was included in the new version of the manuscript.

In results, during the microdialysis in short term experiment, it is not clear if the baseline values ​​of animals treated with CBD (either 50, 100 or 200) before TBI were similar to the SHAM group

Response: We appreciate the comment. The concentration of glutamate under basal conditions of the different experimental groups was not significantly different when compared with the values obtained from the Sham group (see table below). We have included this idea in the new version of the manuscript.

F (4, 30)=0.5458, p=0.7034 according to One-way ANOVA followed by Tukey’s test.

Reviewer 4 Report

The paper entitled “Cannabidiol reduces short- and long-term high glutamate release after severe traumatic brain injury and improves functional recovery,” written by Cindy Santiago‐Castañeda et al. evaluates the effects of cannabidiol (CBD) administration on the short- and long-term extracellular concentration of glutamate and motor dysfunction on male Wistar rats, subjected to Traumatic brain injury (TBI). I think the topic of the manuscript is attractive and clinically helpful. It contains results allowing the authors to suggest that CBD could prevent glutamate‐induced excitotoxicity and functional deficits evoked by severe TBI in humans. Although the study includes a lot of work, the most critical observation is about the rationale of the experimental design. It is not clear why the author used the two experimental protocols, short and long-term protocols. What would be the rationale for testing the effect of CBD in a previous situation of traumatic brain injury? Or even in a pre AND post-injury condition? Why were these doses of CBD selected?. Moreover, the study is based on the measurement of glutamate levels; however, any analytical details are provided in the material and methods section, e.g., how is the detection technique to quantify dialysate glutamate levels?. How authors could confirm that the origin of glutamate is synaptic but not metabolic?. It is critical the author should improve methodological issues and justification to finally accept the manuscript.

Author Response

It is not clear why the author used the two experimental protocols, short and long-term protocols.

Response: We included a rationale for experiment at the beginning of their description.

What would be the rationale for testing the effect of CBD in a previous situation of traumatic brain injury? Or even in a pre AND post-injury condition?

Response: At the end of the introduction section, we included a rationale for testing the effects of CBD pretreatment in TBI. It is also indicate that “We also investigate if the effects of CBD were augmented when is applied pre- and post-TBI.”

The results obtained reveal that pre-treatment with CBD was sufficient to lessen the adverse sequelae of TBI. This finding is described and discussed through the manuscript.

Why were these doses of CBD selected?

Response: A rationale of the doses of CBD selected was included in the Materials section.

Moreover, the study is based on the measurement of glutamate levels; however, any analytical details are provided in the material and methods section, e.g., how is the detection technique to quantify dialysate glutamate levels?.

Response: The new version of the manuscript includes a more detailed description of the HPLC procedure.

How authors could confirm that the origin of glutamate is synaptic but not metabolic?.

Response: We appreciate the reviewer´s comment. Glutamate is synthesized from glutamine via glutamine synthetase. Its metabolism mediated by the mitochondrial complex 2-oxoglutarate dehydrogenase (OGDHC) results in 2-oxoglutarate. Studies indicate that TBI induces inactivation of OGDHC (Mkrtchyan, et al., 2018). This effect can result in enhanced levels of glutamate. In the short term, the increase in glutamate concentrations induced by severe TBI is mediated by the rupture of cell membranes, increased synaptic release and neuronal death. The quantification of glutamine and the estimation of the glutamate/glutamine index can support the synaptic origin of glutamate. Concerning metabolically derived glutamate, enzymatic assays can be performed to estimate OGDHC activity. We expect that orally administered CBD prevents the TBI-induced inactivation of OGDHC and the high extracellular levels of glutamate. The results obtained in the present study are insufficient to determine the origin of glutamate. However, future studies can be carried to support this idea.

It is critical the author should improve methodological issues and justification to finally accept the manuscript.

Response: We appreciate the comment. We improved the methodological section.

Round 2

Reviewer 1 Report

For the figure, please read the conditions.

Reviewer 4 Report

I appreciate the work of the authors. I think that they significantly improved the manuscript after the revision by reviewers.